# Heparin Resistance in Cardiac Surgery with Cardiopulmonary Bypass: Mechanisms, Clinical Implications, and Evidence-Based Management

**DOI:** 10.3390/medicina61122088

**Published:** 2025-11-23

**Authors:** Karina E. Rivera Jiménez, Yahaira M. Mamani Ticona, Giancarlo Gutierrez-Chavez, Cristian O. Astudillo, Edisson Calle, Giancarlo A. Torres Heredia, Dario S. Lopez Delgado, Oriana Rivera-Lozada, Joshuan J. Barboza

**Affiliations:** 1Instituto Nacional Cardiovascular, Lima 15072, Peru; edit.rj@outlook.com; 2Sociedad Científica de Estudiantes de Medicina de la Universidad Nacional del Altiplano (SOCIEM UNA), Puno 21001, Peru; yamamaniti@est.unap.edu.pe; 3Facultad de Medicina Humana, Universidad Nacional del Altiplano, Puno 21001, Peru; 4Facultad de Ciencias de la Salud, Universidad Continental, Cusco 08000, Peru; 5Hospital San Sebastián, Cuenca 010111, Ecuador; 6Facultad de Ciencias Quimicas y de la Salud, Universidad Tecnica de Machala, Machala 070205, Ecuador; edisson.calletorres@gmail.com; 7Facultad de Ciencias de la Salud, Universidad Privada Antenor Orrego, Trujillo 13008, Peru; giancatohe99@gmail.com; 8Hospital Universitario Departamental de Nariño, Pasto 520001, Colombia; 9Vicerrectorado de Investigacion, Universidad Señor de Sipan, Chiclayo 14002, Peru

**Keywords:** heparin resistance, unfractionated heparin, cardiopulmonary bypass, activated clotting time, antithrombin, bivalirudin, argatroban

## Abstract

*Background*: Unfractionated heparin (UFH) is the standard anticoagulant during cardiopulmonary bypass (CPB). A clinically relevant subset develops heparin resistance (HR)—failure to reach adequate anticoagulation with usual UFH—raising thrombotic risk and complicating perioperative care. *Objectives*: To synthesize contemporary evidence on the mechanisms, clinical implications, and perioperative management of HR in adult cardiac surgery with CPB. *Methods*: This narrative review synthesizes contemporary evidence on the epidemiology, mechanisms, recognition, and management of HR in adult cardiac surgery with CPB, emphasizing clinically actionable points. *Results*: Incidence varies across centers and definitions. Mechanisms include antithrombin (AT) deficiency or consumption and AT-independent drivers such as systemic inflammation or sepsis, protein-loss states, thrombocytosis, hyperfibrinogenemia, obesity, prior heparin exposure, and drug interactions. Sole reliance on activated clotting time (ACT) may misestimate anticoagulant effect; anti–factor Xa (anti-Xa) assays or heparin titration systems improve assessment when available. Management is stepwise: UFH dose escalation; targeted AT supplementation (or fresh frozen plasma where concentrates are unavailable); and transition to direct thrombin inhibitors when HR persists or UFH is contraindicated. Protocolized pathways and multidisciplinary coordination reduce delays and adverse events. *Conclusions*: HR is a multifactorial, common challenge in CPB. Pre-bypass risk assessment, multimodal monitoring, and an algorithm prioritizing UFH optimization, AT repletion, and timely use of direct thrombin inhibitors provide a pragmatic framework to limit thrombosis and bleeding. Harmonized definitions and comparative trials remain priorities.

## 1. Introduction

Unfractionated heparin (UFH) is the most widely used anticoagulant in cardiac surgery because of its availability, low cost, straightforward monitoring, and rapid reversal; it remains the first-line drug globally [1]. Heparin anticoagulation enabled cardiopulmonary bypass (CPB) by allowing blood diversion to an extracorporeal circuit with safe oxygenation and perfusion while preventing thrombosis on non-endothelialized surfaces [2]. Initiating CPB requires a systemic UFH dose, a target activated clotting time (ACT), and continuous monitoring to maintain a safe threshold [3]. After separation from CPB, reversal with protamine—and the associated bleeding risk—demands meticulous attention [2].

Up to 30% of patients may exhibit heparin resistance (HR), defined as difficulty achieving the target ACT despite an appropriate UFH dose; repeated dosing with minimal response complicates management, and preventing circuit thrombosis remains paramount [2]. HR is multifactorial, influenced by congenital antithrombin deficiency, prior UFH exposure, systemic inflammatory diseases, prothrombotic states, platelet abnormalities, obesity, and advanced age [4]. CPB further activates coagulation and platelets and, together with hemodilution and longer bypass duration, can blunt UFH’s anticoagulant effect [5]. Although perfusionists often recognize HR first, all team members should understand its risk factors and pathophysiology and enforce rigorous monitoring to improve outcomes [4,6].

This review synthesizes current evidence on HR in cardiac surgery with CPB—definitions and recognition in the operating room, risk factors across preoperative, intraoperative, and circuit-related domains, and therapeutic strategies, including direct thrombin inhibitors (DTI) when target ACT cannot be achieved despite appropriate dosing—translating disparate findings into practical guidance and outlining priorities for research and protocol development.

### Objective

To synthesize current evidence on the epidemiology, mechanisms, recognition, and practical management of HR in adult cardiac surgery with CPB, and to propose an evidence-informed, clinically applicable framework for diagnosis and treatment.

## 2. Methods

### 2.1. Study Design and Objectives

We conducted a narrative review to synthesize determinants, pathophysiological mechanisms, and clinical management strategies of HR in the setting of CPB. Objectives were to: (i) describe contemporary definitions and monitoring approaches; (ii) summarize prevalence and risk factors; and (iii) appraise therapeutic strategies, including alternatives to UFH. A quantitative synthesis was not attempted because HR definitions, monitoring modalities (ACT, anti–factor Xa (anti-Xa), heparin titration systems), and reported outcomes are highly heterogeneous, precluding valid pooling.

### 2.2. Information Sources and Search Strategy

We searched PubMed, Scopus, Google Scholar, and ClinicalTrials.gov for studies published from inception to 31 December 2024 focusing on peer-reviewed studies, using a structured strategy combining “heparin resistance”, “cardiopulmonary bypass”, “anticoagulation”, “cardiac surgery”, and “antithrombin” with Boolean operators. Reference lists of included articles and key reviews were screened for additional records. No language or geographic restrictions were applied.

### 2.3. Eligibility Criteria

Inclusion: Human studies in cardiac surgery with CPB that reported any of the following—definitions/diagnostic criteria for HR, incidence/prevalence, predictors/risk factors, monitoring strategies (e.g., ACT-based or alternative assays), management approaches, and clinical or safety outcomes. We prioritized clinical studies (randomized or non-randomized), cohorts, registries, and high-quality reviews with contemporary relevance and peer review.

Exclusion: Preclinical/animal studies; isolated case reports and letters; studies centered on extracorporeal therapies outside cardiac surgery (e.g., extracorporeal membrane oxygenation [ECMO], ventricular assist devices); and articles lacking sufficient information on HR definitions, predictors, or management strategies.

### 2.4. Study Selection

Two reviewers (KERJ, YMMT) independently screened all titles/abstracts and full texts and performed data extraction using a standardized form. Any discrepancies were resolved by discussion and, when necessary, in consultation with a senior reviewer (DSLD). Additional authors (GGC, COA, EC, GATH, ORL, JJB) supported the development of the search strategy, checked a sample of extracted data for accuracy, and provided methodological oversight.

### 2.5. Data Extraction and Items

We extracted: bibliographic data; setting/design; sample size and patient characteristics; CPB details (duration, hemodilution/hemofiltration practices); operational definition/diagnostic criteria for HR (including ACT targets and UFH dosing thresholds); monitoring modality; HR frequency; identified predictors; management strategies (UFH dose escalation, AT supplementation, DTI, adjunctive measures) and response rates; and safety/clinical outcomes (bleeding, thrombotic events).

### 2.6. Outcomes of Interest

Primary outcomes were operational definitions of HR and its frequency during cardiac surgery with CPB. Secondary outcomes were: (i) patient-, procedure-, and assay-related predictors; (ii) performance of monitoring strategies; and (iii) effectiveness and safety of therapeutic approaches, including alternatives to UFH.

### 2.7. Synthesis Approach

Thematic organization (epidemiology, mechanisms, recognition/monitoring, management) with a focus on actionable guidance for anesthesiologists, surgeons, and perfusionists. No registered protocol; PRISMA not applicable to this narrative review. Quantitative synthesis was not feasible due to major clinical/methodological heterogeneity—HR definitions/ACT targets, monitoring modalities, study designs/comparators, interventions, and outcome measures—and insufficient extractable data.

### 2.8. Quality Appraisal and Reporting

We performed design-appropriate qualitative appraisal: randomization/allocation and completeness of follow-up for trials; confounding/selection, outcome ascertainment, and missing data for observational studies; clarity, scope, and consensus process for guidelines/consensus statements. Appraisal informed interpretation but no numeric scores were assigned, and no studies were excluded solely on rating.

## 3. Results

### 3.1. Prevalence of Heparin Resistance

The prevalence of HR varies widely in the literature, with figures ranging from 4% to 30.7% in adult cardiac surgery populations [7,8]. This marked variability is primarily due to the lack of a clear and universally accepted definition. This complexity, which may reflect the variability in individual patients’ physiological response to heparin, suggests that a single management approach may not be feasible in clinical practice [9]. Table 1 summarizes key studies in this review.

### 3.2. Definition of Heparin Resistance

A survey of members of the Intensive Care and Perioperative Thrombosis and Hemostasis Committee of the International Society on Thrombosis and Haemostasis offers a general definition of HR as a heparin requirement > 35,000 IU/day [14]. Another study specifies that HR may be defined by the inability to raise activated partial thromboplastin time (aPTT) and ACT into their desired therapeutic ranges [15].

In CPB, HR is often defined as failure to achieve a target ACT ≥ 480 s after administering a weight-based dose of 500 U/kg of heparin [6,7,8]. Some studies define HR as the inability to reach the target ACT even after an additional heparin dose, as reported in a descriptive survey of the Japanese Society of Extracorporeal Technology in Medicine [4]. A retrospective review of 756 CABG patients used three criteria to define HR: (1) ACT < 400 s after a 300 IU/kg bolus; (2) ACT < 480 s after ≥ 400 IU/kg; or (3) HSI < 1.3, where HSI = (post-heparin ACT − baseline ACT)/total loading dose (IU·kg^−1^) [16].

Reaching consensus on a standard definition has been difficult because both the typical UFH bolus (300–600 U/kg) and the ACT target (400–480 s) vary across institutions [7]. Some centers use mini-CPB circuits, target ACT 300 s, and administer reduced heparin doses (150 IU/kg) [17]. A review notes that many institutions safely target ACT ≥ 350 s and a heparin level ≥ 2.0 U/mL without reported complications [9,18].

Accordingly, for this article, we consider HR to be present when the initial UFH dose exceeds 400 IU/kg and the target ACT of 500 s is not achieved, or when an additional UFH dose must be administered into the extracorporeal circuit after CPB has begun.

### 3.3. Mechanisms of Heparin Resistance

The pathophysiology of HR is complex and multifactorial, with mechanisms broadly grouped into AT-dependent and AT-independent factors. UFH acts by activating AT, thereby enhancing inactivation of key coagulation factors, including thrombin (IIa) and factor Xa [19,20]. HR is not invariably linked to AT deficiency; a recent retrospective study found that AT deficiency was not a significant risk factor for HR in cardiovascular surgery [10].

In Japan, one study reported an HR prevalence of 89.8%, with 75% associated with AT activity ≥ 80% [4]. More recently, no association was observed between preoperative AT-III activity and ACT after the first heparin dose for CPB, even across varying heparin doses. Rather, higher initial UFH doses tended to yield higher ACT values irrespective of AT-III activity [21].

#### 3.3.1. AT Deficiency

AT is a serine protease inhibitor that inactivates thrombin and other factors. Heparin, by binding and activating AT, accelerates the AT–thrombin reaction ~1000-fold, thereby inhibiting the coagulation cascade [22,23]. Because heparin exerts its effect by catalyzing AT’s anticoagulant activity, AT deficiency has been extensively investigated as a principal factor associated with HR, and AT supplementation is the recommended first-line treatment when AT-dependent HR is confirmed in the laboratory [22,23].

Mean AT activity in adults ranges from 80% to 120%, and deficiency is typically defined as AT activity < 80% [8]. Both acquired and congenital AT deficiency are associated with increased thrombosis risk. In cardiac surgery, AT activity has been reported to be below 50% at preoperative baseline levels [24]

Hereditary AT deficiency is classified as type I (quantitative; reduced antigen and activity) and type II (qualitative; normal antigen with reduced activity). Type II is subdivided into defects affecting the reactive site, the heparin-binding site, and pleiotropic variants that typically involve heparin binding and protein stability [23,25].

HR often arises from functional AT deficiency (type II), wherein exogenous heparin cannot effectively activate AT, leading to undetectable anti-Xa levels despite normal AT antigen concentrations, as demonstrated in clinical studies [26,27]. Congenital AT deficiency is relatively rare, with an incidence of ~1 in 10,000, and usually follows an autosomal dominant inheritance pattern [28].

Acquired AT deficiency results from conditions that impair synthesis, increase loss, or accelerate consumption, including hepatic dysfunction, prematurity, nephrotic syndrome, chylothorax, inflammatory bowel disease, malnutrition, severe burns, and interventions such as major surgery or CPB. It may also be induced by heparin therapy, asparaginase treatment for ALL, or sepsis-related DIC [24,25].

#### 3.3.2. Inflammatory Response

In general, systemic inflammatory states, such as sepsis, cause HR due to increased acute-phase reactants and inflammatory markers that compete for heparin binding, inhibiting its anticoagulant effects [29,30]. Preoperative albumin levels < 3.8 g/dL and fibrinogen levels > 303 mg/dL have been shown to be independent predictors of HR [6]. This immune response reduces heparin bioavailability, requiring higher doses.

Preoperative infective endocarditis has been identified as an independent risk factor for developing HR (Odds Ratio: 4.57), reported as a mechanism not mediated by AT [13,31]. Furthermore, the hypercoagulable state observed in patients with COVID-19, exacerbated by comorbidities such as obesity and diabetes, increases this heparin resistance [30]. In these systemic inflammation scenarios, predicting the response to HNF becomes more complex, making close monitoring essential and requiring continuous dose adjustments [5].

#### 3.3.3. Platelet Count

Elevated platelet counts and fibrinogen levels have been identified as predictors of HR [32]. Thrombocytosis in conditions such as essential thrombocythemia can drive HR because increased platelets potentiate thrombin generation, requiring higher heparin doses to achieve anticoagulation [33]. In addition, cardiac surgery and extracorporeal life support can cause platelet activation and dysfunction, diminishing heparin’s anticoagulant effect [34]. Elevated fibrinogen levels observed in inflammatory conditions may further contribute to HR [29].

#### 3.3.4. Protein Loss

Protein-losing conditions can lead to AT deficiency. This relationship is evident in disorders such as primary intestinal lymphangiectasia, an enteropathy with loss of serum proteins and AT that creates a prothrombotic milieu and may alter the response to heparin. Loss of AT due to proteinuria can also produce HR, since AT is essential for heparin’s anticoagulant effect [35].

Multiple studies identify hypoalbuminemia as an independent risk factor for HR [8]. Albumin concentrations ≤ 35 g/L have been associated with increased HR. Albumin shares structural features with heparin and appears to exert heparin-like activity; its highly negative charge may bind positively charged groups on AT [5]. Patients with nephrotic syndrome may develop paraproteinemia, one of the non–AT-mediated causes of HR. Two case reports described HR not mediated by AT, underscoring the complexity of anticoagulation management in such patients [36].

#### 3.3.5. Obesity

Obesity is linked to a hypercoagulable state characterized by increased thrombotic activity and resistance to fibrinolysis, which can contribute to HR; thromboelastography often shows elevated maximum amplitude, reflecting enhanced platelet function and a prothrombotic tendency [37].

Morbid obesity influences HR through increased volume of distribution, altered pharmacokinetics, and changes in hemostatic parameters. Adipose tissue may also affect heparin binding and clearance, leading to suboptimal anticoagulation. Careful monitoring and potential dose adjustments are required in obese patients receiving therapy [38].

During CPB, obesity necessitates an adjusted heparin-dosing model. Using total body weight for dosing can yield excessive plasma heparin concentrations with overdose and bleeding risk, whereas an ideal body weight-based model targeting a heparin level of 4.5 IU/mL demonstrated that individualized anticoagulation can mitigate obesity-related HR risks [39].

#### 3.3.6. Preoperative Anticoagulant Use

Preoperative use of UFH or low-molecular-weight heparin (LMWH) is associated with an increased risk of recurrent heart failure [40], probably due to the increased heparin requirements from AT consumption, especially when administered preoperatively for myocardial revascularization. One study reported that patients receiving LMWH had a significantly higher risk of recurrent heart failure than those who did not (Odds Ratio 4.8) [41,42]. Another study found an incidence of HR of 8.06% after preoperative heparin administration in patients undergoing cardiac surgery [11].

Thrombotic occlusion and urgent CPB circuit replacement have been reported in a Stanford type A aortic dissection patient after andexanet alfa administration to reverse edoxaban’s anticoagulant effect [42,43]. Andexanet alfa—a modified recombinant protein used to reverse factor Xa inhibitors—can induce HR; AT administration has been successful in such cases [44]. In this regard, it is worth noting that current clinical guidelines, from Anesthesiologists, Cardiovascular Surgeons, and Perfusionists, do not fully address andexanet alfa-induced HR, highlighting the need for updated recommendations [45].

#### 3.3.7. Biochemical and Hematologic Factors

Non–AT-mediated resistance involves complexes with heparin-binding proteins that alter heparin pharmacokinetics/dynamics [8]. Albumin is a major heparin-binding protein; hypoalbuminemia can reduce heparin bioavailability, necessitating higher doses to achieve therapeutic anticoagulation [6]. Elevated serum paraproteins can also cause HR by inhibiting the heparin–AT interaction; a case during CPB with high IgM kappa paraprotein required three times the usual UFH dose to achieve adequate anticoagulation, illustrating nonspecific paraprotein binding to heparin and perioperative risk [46]. UFH pharmacokinetics are strongly influenced by extensive binding to plasma proteins such as platelet factor 4, histidine-rich glycoprotein, and various lipoproteins; these stable complexes modulate bioavailability and half-life. Heterogeneity of protein–heparin complexes, together with interindividual variability in protein levels, yields a highly variable and unpredictable anticoagulant response [5]. Figure 1 shows the principal risk factors of HR.

### 3.4. Clinical Considerations

Identifying preoperative risk factors is crucial for predicting the development of HR and planning management. In the pediatric population, a nomogram model demonstrated high predictive accuracy (AUC-ROC 0.87) for HR, using key preoperative variables such as antithrombin activity, platelet count, and fibrinogen [47].

In the adult population, we lack predictive models, but many risk factors have been studied, such as platelet count, fibrinogen, D-dimer, creatinine, and C-reactive protein (CRP), which were significantly higher in patients who developed HR. It has been concluded that chronic aortic dissection, chronic obstructive pulmonary disease, and smoking, along with elevated fibrinogen, are independent predictors of HR [35]. Other associated factors include elevated factor VIII, age over 55 years, and preoperative heparin treatment [31]. The relevance of these findings lies in the fact that the preoperative evaluation of these factors allows the surgical team to anticipate an inadequate response to heparin and begin planning alternative strategies before the start of CPB.

### 3.5. Anticoagulation Monitoring Systems in CPB

#### 3.5.1. Activated Clotting Time

ACT is considered the standard method to monitor anticoagulation during cardiac surgery with CPB; target values to initiate and maintain anticoagulation during CPB are typically ≥480 s [3]. Advantages include low cost, simplicity, rapid turnaround, a fairly linear relationship with heparin concentrations > 1 U/mL, and the ability to detect heparin effects even at the high concentrations required for cardiac surgery (2–10 U/mL), which would otherwise render aPTT uninterpretable [5].

On the other hand, ACT can yield false results, especially in conditions such as antiphospholipid syndrome [48]. Patient factors (preoperative medications, platelet count/function, coagulation factor deficiencies, proinflammatory states), CPB-related factors (hemodilution, hypothermia), monitoring device characteristics, and the anticoagulant itself all influence ACT results [5].

#### 3.5.2. Hepcon Hemostasis Management System

Hepcon quantifies whole-blood heparin concentration via heparin/protamine titration. A retrospective study of 43 patients found no significant differences in bleeding or thrombotic complications when comparing Hepcon-guided therapy with standard anticoagulation in cardiac surgery [48]. Its utility lies in providing additional information on heparin concentration, useful in cardiac surgery, with good correlation to other devices and laboratory anti-Xa assays [49].

#### 3.5.3. Anti-Xa

Anti-Xa monitoring is increasingly used because of stable anticoagulation control; one study showed better correlation with heparin dosing and fewer blood product transfusions compared with ACT-based management [50]. It is frequently used to monitor anticoagulation during ECMO and with ventricular assist device [51].

#### 3.5.4. aPTT and PT

aPTT and PT are employed to assess heparin anticoagulation during ECMO or ventricular assist device support, but not for CPB [52].

#### 3.5.5. Viscoelastic Tests

Viscoelastic testing evaluates coagulation in cardiac surgery, generally after CPB, to identify causes of bleeding; it can reveal residual circulating heparin that was not adequately reversed [53].

### 3.6. Clinical Management Strategies

#### 3.6.1. Heparin Dose Adjustment

Increasing the heparin dose is a common first-line response to HR [4]. This approach aims to reach the target ACT despite initial resistance [8]. However, only a modest ACT rise—or no response—may occur, as in a case report in which a patient required three times the usual UFH dose to maintain adequate anticoagulation during CPB due to elevated serum IgM-κ paraprotein causing HR [46]. Other studies also describe administering a third heparin dose when the target ACT is not achieved [8].

Intraoperative monitoring of blood heparin levels during CPB is essential because large UFH doses are administered to prevent thrombosis. One study recommends precise methods that evaluate parameters such as ACT, α-angle, and maximum amplitude to ensure effective anticoagulation and global hemostasis throughout the procedure [54]. Nevertheless, this small-volume whole-blood technology is not widely available in many institutions.

#### 3.6.2. Antithrombin Supplementation

AT supplementation is widely used, particularly when HR is associated with low AT activity [8,33]. Multiple studies have employed AT to resolve HR effectively, including in patients with normal baseline AT activity [55]. Compared with fresh frozen plasma (FFP), AT supplementation has been associated with reduced mortality and shorter ICU stays in patients undergoing CPB [20].

The 2024 European Association for Cardio-Thoracic Surgery, European Association of Cardiothoracic Anaesthesiology and Intensive Care, and European Board of Cardiovascular Perfusion (EACTS/EACTAIC/EBCP) guideline for CPB in adult cardiac surgery recommends AT concentrate as primary therapy for AT deficiency to improve heparin responsiveness (Class I, Level B). If AT concentrate is unavailable, FFP should be considered to treat AT deficiency and improve heparin responsiveness (Class IIa, Level C) [18].

Even in cases where HR is induced by agents such as andexanet alfa, AT administration has achieved adequate anticoagulation [44].

#### 3.6.3. Alternative Anticoagulants

When HR occurs, alternative agents such as the DTI bivalirudin and argatroban can be used, offering a viable option when heparin is ineffective or contraindicated [8]. Resistance to these alternatives can also occur, as illustrated by a cardiac-transplant case reportedly exhibiting bivalirudin resistance [56].

Bivalirudin. A reversible DTI with onset 2–4 min and half-life 25–36 min; clearance is predominantly proteolytic, with <20% renal elimination. It does not require AT and is non-immunogenic [57]. Bivalirudin has shown efficacy and safety comparable to heparin for CPB anticoagulation—particularly in HR—with lower rates of mortality and major bleeding compared with heparin cohorts [18]. Successful use has been reported across a wide range of CPB procedures (cardiac transplantation, ventricular assist devices implantation, valve repair), often using an infusion of ~2 mg/kg/h and targeting ACT > 400 s or ≥2.5× baseline [57].

Argatroban. A DTI described in percutaneous procedures, ECMO, ventricular assist devices, and cardiac surgery with CPB under various protocols; in one report, after sternotomy, a 200 μg/kg loading dose achieved ACT > 400 s, followed by continuous infusion starting at 3.5 μg/kg/min and then titrated to 3 μg/kg/min to maintain the desired ACT [58]. A case study reports mitral valve replacement surgery in a patient with heparin-induced thrombocytopenia (HIT) where Ecarin Clotting Time (ECT) monitoring was unavailable. Argatroban was initiated at 2.5 μg/kg/min, without adding a bolus to the circuit priming, achieving an ACT of 500. However, upon declamping, thrombi formed in the circuit, and it had to be replaced with an emergency circuit. The study concludes that, lacking an antidote for its reversal, careful dose management is required, especially during prolonged CPB times [59].

Tirofiban. Considered as an alternative in a case series where bivalirudin could not be used due to elevated creatinine and other DTI were unavailable; the RESTORE protocol (bolus 10 μg/kg, infusion 0.15 μg/kg/min) achieved a target ACT of 480 s in 10 patients [60].

### 3.7. Biocompatibility CPB (Extracorporeal Circulation) and MiECC (Mini-Extracorporeal Circulation)

Required oxygenation flow, extent of biomaterial exposure, and surgery-related tissue trauma trigger coagulation cascades. Immediate blood contact with extracorporeal-circuit biomaterials activates platelets and complement, producing a systemic inflammatory response with cytokine release, endothelial activation, and migration of activated macrophages to peripheral tissues [61].

A study of 68 patients undergoing CABG with MiECC using heparin–albumin–coated circuits and systemic heparin 300 IU/kg (target ACT 300 s) versus a control group (target ACT 450 s) showed a simultaneous decrease in endogenous thrombin potential (ETP) in both groups; no thromboembolic complications occurred. The authors concluded that MiECC with a reduced anticoagulation strategy appears feasible, though larger clinical trials are needed to establish safety [62].

In a cohort of 40 patients undergoing isolated CABG—20 HS and 20 MiECC—no significant differences were observed in CPB time or postoperative transfusion needs. HS showed more favorable biochemical parameters (e.g., lactate, troponin), suggesting outcomes comparable to MiECC with potential biochemical advantages [63]. Heparin-coated MiECC circuits carry a class I, level B recommendation in recent practice guidelines to reduce blood loss and hemodilution and to improve hemocompatibility [18].

Not all studies concur: in a retrospective series of 300 consecutive adults undergoing cardiac surgery with CPB and receiving one of three coatings (Phisio^®^, Trillium^®^, Xcoating™), the heparin required to maintain ACT > 400 s during CPB did not differ by coating choice [64]. Table 2 summarizes management, monitoring and alternatives to heparin.

### 3.8. Critical Appraisal of the Evidence

Across the reviewed studies, there is substantial variability in how HR is defined and managed, which hampers comparability and synthesis of results. A standardized definition and approach would improve the reliability of findings. Although many institutions target ACT 480 s during CPB, in practice targets and systemic UFH dosing vary widely [14]. The diminished response to heparin—commonly labeled HR—still lacks consensus in both diagnosis and treatment [3]. This heterogeneity in diagnostic criteria, ACT targets, and heparin-sensitivity assays further complicates assessment of study quality [8].

Differences in patient populations and settings (e.g., COVID-19) add variability [5]. Other conditions—antiphospholipid syndrome and HIT—may also arise during CPB and confound management despite their autoimmune nature [65]. Regarding treatment strategies, AT supplementation and alternative anticoagulants have been reviewed, but effectiveness is mixed [55]. Finally, the impact of HR on surgical outcomes (thromboembolic events, length of stay) is inconsistently reported, limiting definitive conclusions [4]. Figure 2 summarizes the key mechanisms, confirmation criteria, and management pathways of HR.

Because included studies employed divergent operational definitions (varying ACT thresholds and UFH dose caps), heterogeneous monitoring modalities, and non-uniform outcome measures, a quantitative synthesis would risk misleading precision. Accordingly, we present a structured narrative synthesis to preserve clinical interpretability.

## 4. Discussion

HR during CPB is common and clinically consequential. Across heterogeneous reports, two themes are consistent: (i) ACT alone incompletely reflects anticoagulant effect, and (ii) management is most effective when protocolized—beginning with UFH optimization, proceeding to targeted AT repletion when deficient/consumed, and reserving DTI when HR persists or UFH is contraindicated. These observations align with contemporary professional guidance from cardiac-anesthesia and perfusion societies and with accumulating observational data rather than definitive trials [18].

Congenital AT deficiency is rare; acquired deficiency—type I (quantitative) or, more often, type II with heparin-binding site defects—is the mechanism most linked to HR [26]. Yet many HR cases present with normal or even elevated preoperative AT activity, indicating HR frequently occurs without measurable AT deficiency [10,66]. AT-independent drivers include highly inflammatory states (endocarditis, sepsis, cardiogenic shock, hepatic/multiorgan dysfunction, COVID-19) [35], hypoalbuminemia altering UFH pharmacokinetics/bioactivity [13,21,57], and thrombocytosis (with a prediction model incorporating AT activity, platelet count, and fibrinogen) [47]; chronic thoracic aortic dissection, smoking, and COPD have also been reported [13].

Additional procoagulant influences are type 2 diabetes mellitus, insulin resistance, obesity, advanced age, and preoperative UFH/LMWH therapy [37,42]. Use of andexanet alfa to reverse factor-Xa inhibitors has been associated with HR in this setting [67]. UFH binding to circulating proteins, cells, and non-endothelial surfaces likely reduces bioavailability and contributes to HR [10], and the coexistence of AT-dependent and AT-independent mechanisms underscores management complexity [8,14]. During CPB, ACT remains the main monitoring tool but may overestimate heparin effect under hypothermia or hemodilution, risking underdosing and intraprocedural thrombosis.

In clinical practice, a standardized HR approach is advisable. Teams should (1) screen pre-bypass risk (e.g., prior UFH exposure, inflammatory states, high fibrinogen/platelets, protein-loss conditions); (2) apply structured intraoperative workflows for UFH escalation, targeted AT repletion (or FFP if AT concentrate is unavailable), and timely conversion to DTI (bivalirudin/argatroban); (3) adopt multimodal monitoring (ACT plus anti-Xa or titration systems) when feasible; and (4) document triggers and responses to support post-case learning and quality improvement. Embedding these steps in detailed, protocolized pathways—such as those described by Sharma et al.—can enhance safety and efficacy in HR [57,67]. Consistent with contemporary adult CPB guidance, MiECC circuits—which may limit thrombin generation—should be considered by trained teams, acknowledging limited availability in some regions [18,68,69,70]. Therapeutic thresholds and monitoring modalities should align with current EACTS/EACTAIC/EBCP recommendations used in the manuscript.

Limitations: Heterogeneity in HR definitions, ACT targets, monitoring modalities (ACT vs. anti-Xa vs. titration systems), interventions (UFH escalation, AT repletion, DTI), and outcome measures (bleeding definitions, thrombosis ascertainment, transfusion thresholds)—together with the predominance of non-randomized designs and incomplete reporting—precluded a credible quantitative synthesis and increases the risk of publication/selection bias; consequently, external validity is limited and causal inference is not warranted, and the findings should be interpreted as evidence-informed guidance rather than pooled effect estimates. Even with these constraints, the review offers several counterbalancing strengths: a comprehensive search; a transparent, pre-specified thematic synthesis; and, crucially, a pragmatic algorithm adaptable to resource-rich and resource-limited settings that translates the literature into actionable recommendations for anesthesiologists, surgeons, and perfusionists. HR in CPB requires early recognition, multimodal monitoring (ACT complemented by anti-Xa or titration systems where available), and an algorithmic, team-based response prioritizing UFH optimization, selective AT repletion, and timely DTI use. Implementing protocolized pathways can reduce thrombotic and bleeding risk across diverse resource settings. Standardized definitions and prospective comparative studies are priorities to strengthen practice recommendations.

Future directions. Priorities include consensus HR definition and core outcome set; multicenter registries capturing monitoring modality and treatment sequencing; trials comparing AT strategies (concentrate vs. FFP) and UFH-based algorithms vs. DTI-first pathways; evaluation of anti-Xa/titration-guided strategies vs. ACT alone; and cost-effectiveness analyses in diverse resource settings.

Take-home messages: (i) Define consistently: For adult cardiopulmonary bypass, use HR = inability to achieve an ACT ≥ 480 s after 500 U/kg of UFH before or during CPB. (ii) Check anticoagulation, not just ACT: ACT is essential but not perfect; supplement with anti-Xa or titration systems when available. (iii) Treat in stages: Optimize UFH → selectively replace anticoagulant → use FFP if anticoagulant is unavailable → switch to a direct thrombin inhibitor when UFH is ineffective or contraindicated. (iv) Recognize at-risk patients: Infective endocarditis and hypoalbuminemia are frequently associated with HR; anticipate and plan accordingly. (v) Be pragmatic in all environments: Protocolize care; in resource-limited contexts, viable substitutions (e.g., FFP) with careful monitoring can maintain safety.

## 5. Conclusions

HR during CPB is best managed through a protocolized, multidisciplinary algorithm rather than ad hoc dose adjustments. Key components are pre-bypass risk stratification, multimodal anticoagulation monitoring and a structured, stepwise approach to UFH escalation with targeted AT (or FFP) supplementation, timely switch to DTI, and, where feasible, use of minimally invasive extracorporeal circulation. Future research should focus on establishing a consensus definition and core outcome set for HR and on developing multicenter registries and pragmatic, cost-effectiveness-oriented comparative trials to identify the safest and most efficient management strategies.

## Figures and Tables

**Figure 1 medicina-61-02088-f001:**
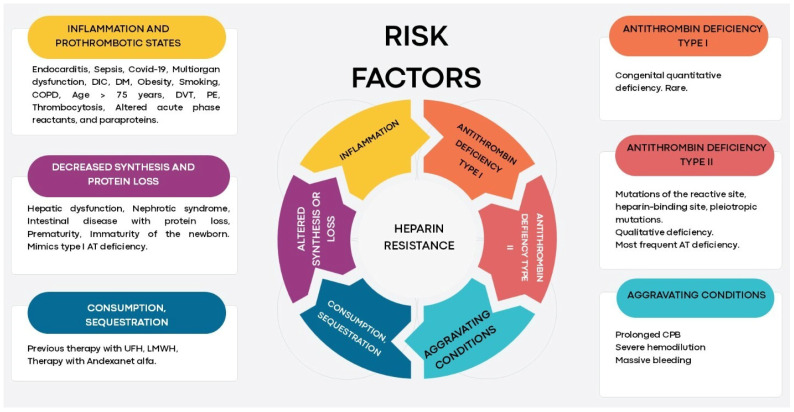
Risk factors associated with heparin resistance. Heparin resistance arises from multiple mechanisms, including proinflammatory and prothrombotic states, congenital or acquired antithrombin deficiency (type I or II), decreased protein synthesis or loss, prior therapy, consumption or sequestration, and perioperative aggravating conditions; these mechanisms can act synergistically to reduce heparin efficacy and increase thrombotic risk. Abbreviations: AT, antithrombin; COVID-19, coronavirus disease 2019; CPB, cardiopulmonary bypass; COPD, chronic obstructive pulmonary disease; DIC, disseminated intravascular coagulation; DM, diabetes mellitus; DVT, deep vein thrombosis; LMWH, low-molecular-weight heparin; PE, pulmonary embolism; UFH, unfractionated heparin. Created by the authors.

**Figure 2 medicina-61-02088-f002:**
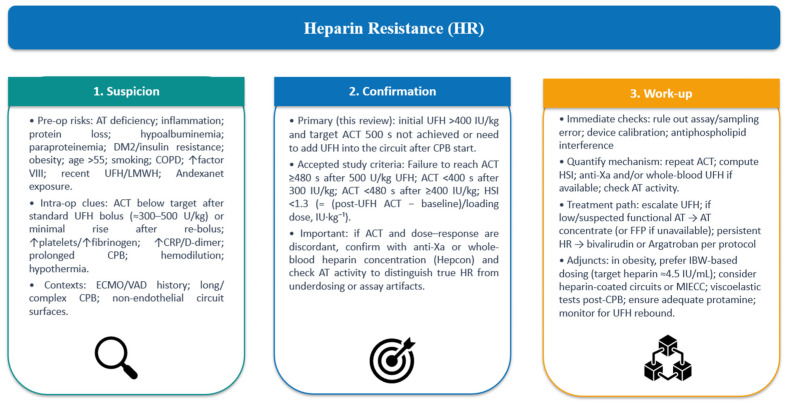
Conceptual framework for HR. Three-tier overview: suspicion (pre-op risks, intra-op clues), confirmation (ACT-based criteria), and work-up (checks; ACT/anti-Xa/Hepcon/AT; UFH escalation, AT concentrate/FFP, DTI; adjuncts and documentation). Abbreviations: HR—heparin resistance; CPB—cardiopulmonary bypass; ACT—activated clotting time; UFH—unfractionated heparin; HSI—heparin sensitivity index; anti-Xa—anti–factor Xa activity; Hepcon—heparin/protamine titration system; AT—antithrombin; FFP—fresh frozen plasma; DTI—direct thrombin inhibitor(s). Created by the authors.

**Table 1 medicina-61-02088-t001:** Summary of key studies on HR in cardiac surgery with CPB.

**First** **Author**	**Study Design**	Population	HR Definition	UFH Dose(s) at Which ACT Target Failed (Initial Bolus)	Main Outcomes	Conclusion
Youngin Kim et al., 2025; South Korea [10]	Retrospective cohort	Valve and aortic surgeries with CPB; emergency aortic dissection excluded; preop antithrombin activity measured ≤ 15 days pre-op; pre-heparin ACT ≤ 180 s; *n* = 605.	ACT < 450 s after initial UFH 3 mg/kg (300 IU/kg) before CPB.	Initial bolus 300 IU/kg; HR labeled if ACT < 450 s after this dose; cumulative before labeling HR: 300 IU/kg.	Major composite complications: 20.1% HR vs. 12.5% non-HR (*p* = 0.025); individual complications not significantly different; ICU stay 47.1 h (IQR 24.7–72.1) HR vs. 46.8 h (24.8–72.2) (*p* = 0.731); hospital stay 11 d (8–19) HR vs. 9 d (7–14) (*p* = 0.011); early death ≤ 30 d: 3.7% HR vs. 3.2% non-HR (*p* = 0.972).	HR risk increases with active infection and isolated aortic surgery; pre-op antithrombin activity is not linked to HR; routine prophylactic pre-op AT supplementation is not indicated.
Butt et al., 2024; United Arab Emirates [8]	Systematic Review	Adults undergoing cardiac surgery with cardiopulmonary bypass; ICU contexts discussed.	Commonly accepted: need for >500 IU/kg UFH to achieve ACT 480 s; HSI < 1 s·kg/IU indicative of HR.	Typical initial bolus 300–400 IU/kg; HR labeled when cumulative requirement exceeds 500 IU/kg to reach ACT 480 s.	HR frequency reported 4–26% depending on initial bolus and ACT target; independent predictors include infective endocarditis and albumin ≤ 3.5 g/dL; management options include additional UFH, FFP, AT concentrates, DTIs (bivalirudin/argatroban), and nafamostat.	Optimize HR diagnosis (ACT/assays) and management via multidisciplinary approach; consider AT supplementation and DTIs when indicated; individualized strategies improve perioperative outcomes.
Jerrold H. Levy et al., 2024; USA [7]	Systematic Review	Adults undergoing cardiac surgery with cardiopulmonary bypass (28 studies; 11,382 patients).	Proposed standardized definition—failure to reach ACT ≥ 480 s after 500 U/kg UFH.	Initial bolus varied across studies (commonly 300–600 U/kg); labeling HR occurred at study-specific cumulative thresholds of 300, 400, or 500 U/kg; authors propose 500 U/kg as the standardized threshold.	Most frequent ACT targets were 480 s (46% of studies), 400 s (21%), and 450 s (18%); most common maximum UFH thresholds were 300, 400, or 500 U/kg; weighted-by-patients, the prevalent combination was ACT 480 s with 500 U/kg; pooled HR incidence 17.5% (1975/11,382).	Standardize HR as failure to achieve ACT ≥ 480 s after 500 U/kg UFH to harmonize research and clinical management; current incidence and practice vary widely.
Koki Ito, 2023; Japan [4]	Cross-sectional survey	Targeted surgical cases with CPB across 537 institutions in 2019; responses from 341 institutions (63.5%). Aggregate volume at responding centers: 37,397 CPB procedures in 2019. Key patient-level baseline traits: Not reported.	Most common: failure to reach the target ACT despite an additional dose of heparin (reported by 69% of institutions). Common ACT targets included 480 s (52%) and 400 s (30%).	Initial dose commonly 300–349 IU/kg (71.8% of institutions). When HR present, total dose judged compatible was ~400 IU/kg in 32% of institutions. Exact bolus/cumulative thresholds per patient: Not reported.	Institution-reported HR observed at 89.8% of respondent institutions. Many centers reported HR even with baseline AT ≥ 80%. Effectiveness of AT concentrate reported by 96% of institutions that used it; ≥70% effectiveness in most of those centers.	HR occurs at many cardiovascular centers, including patients with normal AT activity; AT concentrate often resolved HR regardless of baseline AT activity.
Sniecinski, R., 2019; USA [3]	Cross-sectional survey	Society of Cardiovascular Anesthesiologists members caring for adult CPB; 550 analyzed responses (18.5% response rate), predominantly North America.	Not reported (term used; no single survey definition).	Not reported (most use empiric weight-based dosing; thresholds varied).	74.9% used empiric weight-based UFH; 70.7% targeted ACT 400 or 480 s for CPB initiation; 17.1% reported ACT practices below 2018 STS/SCA/AmSECT guidance; HR encountered in 1–10% of cases by 54.2% of respondents; first-line therapy for HR was AT concentrate in 54.2% vs. FFP in 38.4%.	Most target ACT 400–480 s; >15% report practices outside 2018 guidelines; use of antithrombin concentrate for HR has increased.
Onur Saydam et al., 2019; Turkey [11]	Retrospective cohort	Adults undergoing isolated on-pump CABG; exclusions: redo, off-pump, emergency CABG, combined CABG + valve; further exclusions: malnutrition, liver/renal dysfunction (incl. nephrotic syndrome), known AT deficiency, personal/family thromboembolic history; N = 139.	Target ACT ≥ 400 s to start CPB. For ACT < 400 s after first UFH dose, calculated HSI; HSI < 1.3 considered HR.	Initial bolus 300 IU/kg; if ACT < 400 s → additional 150 IU/kg. HR assessed after first dose when ACT < 400 s.	HR incidence 20.9%; CPB and aortic clamp times longer in HR+; FFP and thrombocyte units higher in HR+; re-exploration 2 patients (LMWH+/HR+); mortality overall 3.6% (5/139), higher in LMWH+/HR+ (4 deaths).	Preoperative LMWH may cause intraoperative HR in on-pump CABG; recommend corrective/preventive management with close follow-up.
Mohamed Tarek Elsayegh et al., 2024 [12]	Cross-sectional study.	Adults >18 y undergoing cardiac surgery with CPB; confirmed COVID-19; N = 229; mean age 55.1 ± 9.6 y; 64.6% male; DM 70.7%; HTN 59%; procedures: CABG 48.9%, MVR 18.3%, DVR 11.8%, AVR 5.2%, others.	ACT < 400 s after 300 U/kg unfractionated heparin.	Initial bolus 300 U/kg; HR assessed at ACT after this dose.	Incidence of HR 9.2%. Other outcomes (thrombosis/circuit clot, bleeding/transfusion beyond FFP, circuit change, mortality, ICU/hospital LOS, complications): Not reported.	HR occurred in 9.2% of adult cardiac surgery patients undergoing CPB in the COVID-19 era.
Kimura et al., 2022; Japan [13]	Retrospective cohort	Adult cardiovascular surgery with CPB; consecutive cases Jan 2012–Sep 2018; N = 287. Procedures: thoracic aortic replacement 34.8%, valvular 56.4%, CABG 8.7%. Key baseline traits: Age ~70–71 y (group medians); % male (by group: 65.3% non-HR, 55.7% HR); overall not reported.	Failure to achieve ACT ≥ 400 s after a single initial UFH dose (protocol initial 300 IU/kg; allowable range 250–350 IU/kg). Target ACT during CPB maintained >480 s.	Initial bolus 250–350 IU/kg (protocol 300 IU/kg). HR labeled if ACT < 400 s after initial bolus; additional UFH up to total 450 IU/kg given after HR classification; CPB circuit primed with 5000 IU.	Incidence of HR 30.7% (88/287). Independent risk factors after adjustment: infective endocarditis (OR 4.57, 95% CI 1.10–19.1), albumin ≤ 3.5 g/dL (OR 3.17, 95% CI 1.46–6.93). Lower HSI in HR and in IE. Ability to initiate CPB achieved after additional UFH/AT-III (explicit success rate not separately reported).	Infective endocarditis and preoperative hypoalbuminemia were independently associated with heparin resistance; patients with IE had lower heparin responsiveness (lower HSI). Further work needed on optimal anticoagulation strategies.

Abbreviations: HR, heparin resistance; UFH, unfractionated heparin; ACT, activated clotting time; CPB, cardiopulmonary bypass; CABG, coronary artery bypass grafting; AVR, aortic valve replacement; MVR, mitral valve replacement; DVR, double valve replacement; AT, antithrombin; FFP, fresh frozen plasma; DTI, direct thrombin inhibitor; LMWH, low-molecular-weight heparin; ICU, intensive care unit; LOS, length of stay; HSI, heparin sensitivity index; IE, infective endocarditis; IQR, interquartile range; MiECC, minimally invasive extracorporeal circulation; Pre-op: preoperative.

**Table 2 medicina-61-02088-t002:** Anticoagulation management, monitoring and alternatives to heparin in cardiac surgery with extracorporeal circulation.

Situation	Recommended Intervention	Class/Level of Evidence	Reference Doses
Standard initial anticoagulation	UFH intravenous.	I/B	Initial bolus of 300–400 IU/kg achieve the ACT target. If customized heparin delivery systems are available, these can be used to adjust heparin doses.
Monitoring target	The ACT must be routinely monitored during Cardiac Surgery.	I/C	ACT target of 480 s (or ACT 300 s in MiECC circuits).
HR management (antithrombin deficiency)	Administer Antithrombin (AT) concentrate, as primary treatment to improve heparin sensitivity.	I/B	Dose of 500–1500 to elevate the AT level to the normal range (guided by AT activity measurement).
HR management (at alternative)	If AT concentrate is not available, the use of FFP should be considered.	IIa/B	Administer dose based on weight (generally 10–15 mL/kg).
alternative anticoagulant (bivalirudin)	Should be considered as first-line anticoagulant treatment in patients with acute HIT type 2 requiring cardiac surgery. HR: If ACT target is not achieved despite exhausting all measures (UFH, AT, FFP), it should be considered as an alternative.	IIa/B	I.V. bolus of 1 mg/kg followed by a continuous infusion of 2.5 mg/kg/h. A 50 mg bolus is usually added to the pump prime. Target ACT is 2.5 times baseline, generally >300 s.
Alternative anticoagulant (argatroban)	In patients with acute HIT type 2 requiring cardiac surgery with CPB and significant renal dysfunction. HR: If ACT target is not achieved despite exhausting all measures (UFH, AT, FFP), it should be considered as an alternative.	IIa/B	I.V. bolus of 0.1 mg/kg, followed by infusion of 5 to 25 µg/kg/min (rate can be increased up to 40 µg/kg/min. A 4 mg bolus may be added to the prime. Monitoring preferably via Ecarin Clotting Time (ECT), although ACT targets 450–500 s have been described.
Biocompatibility	Use heparin-coated CPB circuits to reduce blood loss, hemodilution, and increase hemocompatibility.	IIa/B	Not applicable.
Use of MiECC circuits	It is recommended to optimize CPB systems, following a similar approach to MiECC, to reduce blood loss, hemodilution, and increase hemocompatibility.	I/B	Not applicable.

Evidence supported by the 2024 EACTS/EACTAIC/EBCP Guidelines on CPB in adult cardiac surgery [18]. Abbreviations: UFH, unfractionated heparin; ACT, activated clotting time; HR, heparin resistance; AT, antithrombin; FFP, fresh frozen plasma; HIT, heparin-induced thrombocytopenia; CPB, cardiopulmonary bypass; MIECC, minimally invasive extracorporeal circulation; ECT, ecarin clotting time; IV, intravenous.

## Data Availability

The original contributions presented in this study are included in the article. Further inquiries can be directed to the corresponding author.

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
