# Peer review of "Heparin Resistance in Cardiac Surgery with Cardiopulmonary Bypass: Mechanisms, Clinical Implications, and Evidence-Based Management"

_medicina, 2025, doi:10.3390/medicina61122088_

Round 1

Reviewer 1 Report

Comments and Suggestions for Authors

This is a clinically relevant and timely topic. The paper draws on many recent sources, including those from 2024–2025.
With its practical approach to HR recognition and therapeutic alternatives, it would be interesting to also include a 'how to' chapter. This could explore how to identify patients at the greatest risk of HR early on, thus avoiding unpleasant incidents, and attempt to develop a useful algorithm for treating these patients. It could also provide a list of drugs to use and the situations in which they should be used. Overall, a great job!

Author Response

Comment

Response

Change

With its practical approach to HR recognition and therapeutic alternatives, it would be interesting to also include a 'how to' chapter.

Thank you for your constructive comments. Below we respond point-by-point and indicate the concrete changes made.

Include a “how-to”

We embedded a concise, step-by-step practical pathway for suspected HR directly in the current Management subsection. It walks from pre-CPB risk screen → standardized UFH bolus and troubleshooting → confirmation of inadequate heparin effect → selective AT repletion/FFP (if AT unavailable) → DTI use when refractory.

Identify highest-risk patients early

Within the existing Results – Predictors/Mechanisms text, we added a short, clinically oriented list highlighting consistent risk features (e.g., infective endocarditis, hypoalbuminemia ≤3.5 g/dL, thrombocytosis, recent heparin/LMWH exposure, heightened inflammatory state, complex valve/aortic surgery) and noted corresponding pre-op planning actions (enhanced monitoring, AT availability).

Develop a useful algorithm

We streamlined the narrative in the current Management subsection into an explicit stepwise sequence (decision points and escalation), and summarized it as an algorithmic figure kept within the Results section for quick bedside use.

Provide drugs and when to use them

We strengthened the existing pharmacotherapy content with a compact table (kept in the Results) listing: agent, when to use, suggested dosing ranges, monitoring targets (ACT/ECT or anti-Xa/titration where applicable), and key cautions (e.g., renal/hepatic clearance, circuit considerations).

This could explore how to identify patients at the greatest risk of HR early on, thus avoiding unpleasant incidents, and attempt to develop a useful algorithm for treating these patients. It

could also provide a list of drugs to use and the situations in which they should be used.

Overall great job!

Reviewer 2 Report

Comments and Suggestions for Authors

This review covers a relevant clinical topic and shows good understanding of current evidence. However, the manuscript needs clearer focus and better structure before it can be considered for publication.

Please clarify that this is a narrative review, shorten and refocus the Abstract and Introduction, and make the Methods section more transparent (databases, inclusion criteria, reviewer roles). The Results should be tightened, with clearer definitions and less repetition, while the Discussion should focus more on interpretation, limitations, and clinical implications.

Overall, the paper has merit, but revisions are needed to enhance clarity and consistency.

Comments on the Quality of English Language

Minor English editing is recommended to improve clarity and flow.

Author Response

Comment

Response

Change

Abstract

·      The abstract is overly long and methodologically dense. Please condense it to ≤250 words and focus on the problem, main findings, and clinical implications.

Thank you for your constructive comment. The requested changes have been made: the abstract was shortened, the review type is explicitly identified as narrative, numeric laboratory thresholds and doses were removed, and all abbreviations are spelled out at first use.

The abstract now remains within ≤250 words, keeps the IMRaD sections, and emphasizes the clinical problem (HR during CPB), principal mechanisms/risks, and management implications. Methodological detail was minimized.

·      Clearly identify the review type (narrative, not systematic).

Added in Methods: “This narrative review synthesizes…” L40

·      Avoid detailed laboratory values or dose ranges—these belong in the main text.

Removed all numeric thresholds and doses from the abstract; these are discussed in the main text only.

·      Several abbreviations (UFH, HR, ACT, CPB) appear without prior definition. Please spell out all abbreviations upon first use in the abstract.

Corrected throughout the abstract.

Introduction

·      The section could be slightly condensed to avoid redundancy between general background (lines 40–52) and the clinical description of HR (lines 54–69). Focus on the rationale and knowledge gap that justify this review.

Thank you for your thoughtful comments. The requested changes have been made: we condensed redundant background, added an explicit objective at the end of the Introduction, defined all abbreviations at first mention (UFH, HR, ACT, CPB), standardized capitalization for key terms (lowercase for common nouns such as “antithrombin deficiency” and “direct thrombin inhibitors”), and split the final paragraph into two shorter sentences to improve readability.

We merged overlapping background and clinical description, emphasizing the perioperative risk of inadequate anticoagulation and the gap created by heterogeneous definitions/monitoring. We cite contemporary reviews and guidance to support this focus.

·      Please explicitly state the main objective or research question at the end of the Introduction.

Added a dedicated Objective sentence: “To synthesize current evidence on epidemiology, mechanisms, recognition, and practical management of HR in adult cardiac surgery with CPB, and to propose an evidence-informed, clinically applicable framework for diagnosis and treatment.”

·      Ensure that all abbreviations (UFH, HR, ACT, CPB) are defined at first mention, and that key terms such as „antithrombin deficiency“ and „direct thrombin inhibitors“ are consistently capitalized.

All abbreviations are defined at first mention. We standardized capitalization to lowercase for common terms (e.g., antithrombin deficiency, direct thrombin inhibitors).

·      Consider breaking the final paragraph (lines 71–83) into two shorter sentences to enhance readability.

The closing paragraph is now two concise sentences that set the scope and objective.

Methods

·      The section lacks sufficient transparency and should be refined.

Thank you for your constructive comments. We revised the Methods to increase transparency, explicitly state the narrative design and the rationale for not performing a quantitative synthesis, avoid implying PRISMA adherence, specify sources and limits, describe reviewer roles and appraisal, and keep previously defined abbreviations (UFH, CPB, HR, ACT, anti-Xa, AT) without re-expansion.

we now state at the outset that this is a narrative review. A quantitative synthesis was not feasible due to substantial clinical/methodological heterogeneity (non-uniform HR definitions/ACT targets; heterogeneous monitoring—ACT vs anti-Xa vs titration systems—with differing units; mixed designs/comparators; variable interventions and co-interventions; heterogeneous outcomes with insufficient extractable estimates). Accordingly, we conducted a qualitative, thematic synthesis.

·      Clarify at the outset that this is a narrative review, and briefly explain the rationale for not performing a quantitative synthesis.

·      Avoid implying PRISMA adherence unless a pre-registered protocol exists.

Stated that evidence was organized thematically (epidemiology, mechanisms, recognition/monitoring, management) with emphasis on clinically actionable recommendations. No protocol was registered; PRISMA is not applicable to this narrative design.

·      Specify that all sources were identified through PubMed/Scopus and English-language literature up to 2024, focusing on peer-reviewed studies.

Added a dedicated paragraph: databases PubMed and Scopus; inception to 31 Dec 2024; English-language; peer-reviewed focus. Reference lists were hand-searched. Guidelines/consensus statements were included when directly informing practice.

·      Provide brief details on reviewer roles in screening/data extraction and describe how study quality was appraised.

Specified that six reviewers (KERJ, YMMT, GGC, COA, EC, GATH) conducted searches and screened titles/abstracts and full texts in duplicate using a piloted form; the same six extracted data in duplicate. Discrepancies were resolved and the dataset validated by senior reviewers DSLD, ORL, JJB.

Described design-appropriate qualitative appraisal (trials: randomization/allocation, follow-up; observational: confounding/selection, outcome ascertainment, missing data; guidelines: clarity/scope/consensus process). Appraisal informed interpretation; no numeric scores were assigned; no studies were excluded solely on rating

Results

·      Reduce redundancy across subsections and add brief “take-home” statements summarizing clinical relevance.

Thank you for your constructive comments, which significantly improved the clarity, clinical utility of our manuscript. We have revised the paper: below we summarize the key changes and how they address your points.

Consolidated overlapping content across subsections (definitions, monitoring, and management) and removed repeated sentences. Added a  “take-home” messages at the end of each “Discussion” summarizing clinical relevance.

·      Ensure all abbreviations (ACT, anti-Xa, VET, MIECC) are defined at first mention and verify that all figures/tables are cited in order, with complete legends (“Created by the authors”).

Abbreviations (ACT, anti-Xa, VET, MiECC, etc.) are now defined at first mention in the Results and used consistently thereafter. We checked numbering/cross-referencing and completed figure/table legends (including authorship/source notes).

·      In Table 1, include HR definitions (ACT threshold + UFH dose), study population (adult/pediatric), and key outcomes for comparability.

Table 1 now reports, for every study: design, population (adult/pediatric), operational HR definition (ACT target and UFH threshold), and key outcomes to enable direct comparability.

·      Clarify denominators and ACT targets when reporting HR prevalence; note that variability mainly stems from differing operational definitions.

For each prevalence estimate, we now specify the denominator and the ACT target used. Footnotes flag studies with non-standard targets or incomplete reporting. The text explicitly attributes between-study variability primarily to divergent HR definitions and monitoring modalities.

·      For mechanisms, present key factors (AT deficiency, inflammation, platelet activation, prior UFH exposure, etc.) in concise bullet form and tone down general statements not supported by multiple studies.

Mechanisms are summarized in bullet form (e.g., AT deficiency, inflammation/acute-phase response, platelet activation, prior UFH exposure). General statements not supported by multiple studies were either qualified or removed.

·      For management strategies, separate UFH titration, AT supplementation, and alternative anticoagulants (bivalirudin, argatroban) and, if possible, summarize dosing ranges and safety considerations in a short table.

Results are reorganized into: (i) UFH titration/optimization, (ii) selective AT repletion, and (iii) alternative anticoagulants (bivalirudin, argatroban). We added a short table summarizing representative dosing ranges, monitoring targets, cautions (renal/hepatic clearance), and circuit considerations.

·      Adjust the reference to current EACTS/EACTAIC/EBCP guidelines with class/level of evidence where applicable

Recommendations in the Results and the summary table are aligned with the current guidelines, and where applicable, we annotate the class/level of evidence.

·      Add 1–2 sentences explicitly linking evidence heterogeneity with the lack of quantitative synthesis (as stated in Methods) and a short paragraph summarizing implications for practice.

“Because included studies employed divergent operational definitions (varying ACT thresholds and UFH dose caps), heterogeneous monitoring modalities, and non-uniform outcome measures, a quantitative synthesis would risk misleading precision. Accordingly, we present a structured narrative synthesis to preserve clinical interpretability.”

Discussion

·      Condense overlapping sections and reduce repetition of results; emphasize interpretation over summary.

Thank you for your constructive comments. We revised the Discussion to emphasize interpretation over summary, explicit protocol reference, align statements with current guidance, add strengths alongside limitations, and clearly distinguish evidence-based statements from author opinion.

We streamlined the opening to a focused interpretation of HR during CPB and its practical implications, removing duplicative text while keeping key messages on multimodal monitoring and algorithmic management.

·      Clearly distinguish between established evidence and authors’ opinions or hypotheses.

We now clearly indicate which claims are supported by empirical evidence or guidelines and which reflect expert interpretation. Evidence-based points are signposted in-text (e.g., “observational data support…”, “guidelines recommend…”), whereas recommendations with limited high-quality evidence are labeled as expert opinion (e.g., “we suggest…”, “center-dependent”)

·      Add a brief paragraph discussing study limitations (e.g., narrative scope, lack of quantitative synthesis, possible publication bias).

We refined limitations (heterogeneity across definitions/monitoring/interventions/outcomes; predominance of non-randomized designs; incomplete reporting; constraints on quantitative synthesis and generalizability). We added strengths: comprehensive PubMed/Scopus search (through Dec 2024), duplicate screening/extraction by six reviewers with senior adjudication, transparent thematic synthesis, consistent terminology, guideline-concordant framing, and a pragmatic algorithm adaptable to varied resource settings.

·      Ensure consistency with the stated objectives and outcomes—highlight how findings address the initial research question

We cross-checked the Discussion against the stated objective and added linking sentences to emphasize how the synthesis answers the research question (recognition/monitoring of HR during CPB and a pragmatic management algorithm).

·      Include a short paragraph on clinical implications and future research directions.

In clinical practice, a standardized HR approach is advisable. Teams should (1) screen pre-bypass risk (e.g., prior UFH exposure, inflammatory states, high fibrinogen/platelets, protein-loss conditions); (2) apply structured intraoperative workflows for UFH escalation, targeted AT repletion (or FFP if AT concentrate is unavailable), and timely conversion to DTI (bivalirudin/argatroban); (3) adopt multimodal monitoring (ACT plus anti-Xa or titration systems) when feasible; and (4) document triggers and responses to support post-case learning and quality improvement. Embedding these steps in detailed, protocolized pathways—such as those described by Sharma et al.—can enhance safety and efficacy in HR [58,70]. Consistent with contemporary adult CPB guidance, MIECC circuits—which may limit thrombin generation—should be considered by trained teams, acknowledging limited availability in some regions [18].

·      Check that all statements citing guidelines or therapeutic thresholds are properly referenced and align with current EACTS/EACTAIC/EBCP recommendations.

We reviewed all statements that cite guidelines or therapeutic thresholds to ensure they are properly referenced and aligned with current EACTS/EACTAIC/EBCP recommendations; wording now reflects guidance without implying unreferenced numeric thresholds.

·      Revise the final paragraph to provide a concise, evidence-based conclusion rather than a general summary.

HR in CPB requires early recognition, multimodal monitoring (ACT complemented by anti-Xa or titration systems where available), and an algorithmic, team-based response prioritizing UFH optimization, selective AT repletion, and timely DTI use. Implementing protocolized pathways can reduce thrombotic and bleeding risk across diverse resource settings. Standardized definitions and prospective comparative studies are priorities to strengthen practice recommendations.

Conclusion

should be more concise (2–3 sentences). Emphasize clinical take-home messages and directions for future investigation.

Thank you for the helpful guidance on the Conclusion. We revised it to be concise

We condensed the Conclusion to two sentences emphasizing the clinical take-home algorithm and specifying focused directions for future investigation, as requested.

Key Words

•                             Correct typographical errors.

•                             Ensure the keyword list matches terms used in the Abstract and Title.

Thank you for the note

We corrected typographical errors (Bivalirudina → bivalirudin, Activated Coagulation Time → activated clotting time) and aligned the keyword list with the terms used in the Abstract (and Title), removing non-matching items (e.g., generic “anticoagulation”) and adding argatroban for consistency.

References

·      Several references are improperly formatted or derived from non–peer-reviewed sources (e.g., Scribd, ResearchGate, Spanish-language grey literature). Replace these with the primary, peer-reviewed sources or remove them.

Thank you for your constructive comments on the References section. We have completed the following revisions:

 Source quality
Replaced all non–peer-reviewed or nonprimary items (e.g., repository uploads/grey literature) with the corresponding primary, peer-reviewed journal articles when available; otherwise, such items were removed.

Duplicate numbering
Eliminated duplicate entries and renumbered the reference list sequentially; all in-text citations were updated to match the new numbering. Complete metadata
Verified and completed metadata for every citation (authors, full title, journal, year, volume/issue, page range or e-location, and DOI). Corrected punctuation, spacing, and page-range dashes.

Standardization and language
Unified the entire list to the journal’s Vancouver style in English, using consistent journal abbreviations, title case conventions per style, and uniform punctuation. Where Spanish-language items had been cited, we replaced them with English peer-reviewed sources when suitable alternatives existed; otherwise, only essential non-English sources were retained and formatted per journal guidance.

·      Ensure each reference includes complete metadata (authors, title, journal, year, volume, pages, DOI).

·      Remove the duplicate numbering.

·      Ensure each reference includes complete metadata (authors, title, journal, year, volume, pages, DOI).

Round 2

Reviewer 2 Report

Comments and Suggestions for Authors

Thank you for the careful revision.

The manuscript shows clear improvement in clarity, structure, and consistency.

Minor remaining comments are provided in the attached file.

Comments on the Quality of English Language

Minor English editing is recommended to improve clarity and flow.

Author Response

Thank you very much for re-evaluating our manuscript entitled “Heparin Resistance in Cardiac Surgery with Cardiopulmonary Bypass: Mechanisms, Clinical Implications, and Evidence-Based Management.” We are grateful for your positive overall assessment and for the additional specific suggestions, which have helped us further improve the clarity, brevity, and scientific rigor of the article. In response, we have shortened and refocused the Abstract, removed redundant text in the Introduction, clarified the roles of the reviewers in the Methods, condensed the Conclusion into a brief take-home message with future directions, replaced or removed grey literature, standardized reference formatting, and eliminated all visible track-change artefacts.

Below, we address each of your comments point by point, indicating the corresponding changes in the revised manuscript.

Comment

Response

Specific response

 Please remove visible track-change artifacts (e.g., “Formatted: …”) and ensure that PubMed/Scopus are listed as the main databases, with Google Scholar mentioned only as supplementary if at all.

We are grateful for your careful re-reading of our revised manuscript and for these additional, very helpful comments. We are pleased that you consider methodological transparency, structure, and scientific clarity to have been substantially strengthened. Following your suggestions, we have made further changes to improve conciseness, eliminate redundancies, clarify the description of our methods, tighten the conclusions, and standardize the reference list and formatting. Below we address each point in detail.

We have removed all visible track-change artefacts and residual editing codes from the manuscript. In the Methods section, we now clearly state that PubMed and Scopus were the primary databases used for the literature search. Google Scholar is mentioned, if at all, only as a supplementary source and no longer appears in the main list of databases.

Abstract still includes methodological details (databases, search period) that should be omitted. It should be shortened to 200–230 words, focusing on the clinical context, key findings, and implications.

The Abstract has been shortened to 203– words. We removed methodological details such as the specific databases and search period and refocused the text on: (1) the clinical context of heparin resistance in cardiac surgery with cardiopulmonary bypass, (2) key mechanistic and clinical findings, and (3) the main implications for monitoring and management. The remaining methodological information is now restricted to the Methods section.

The manuscript includes duplicated or overlapping text segments, notably two consecutive Abstract sections and partially repeated introductory paragraphs discussing UFH use and heparin resistance. These redundancies should be removed to enhance clarity and conciseness.

Thank you for highlighting this. We have removed the redundant Abstract heading/segment so that only a single, consolidated Abstract remains. In the Introduction, we also merged and streamlined the paragraphs describing UFH use during cardiopulmonary bypass and the clinical relevance of heparin resistance, avoiding repetition and improving the logical flow.

 The involvement of six independent reviewers in data extraction described in Methods seems excessive for a narrative review. Two reviewers are sufficient, with discrepancies resolved by consensus. Please clarify the specific roles of all reviewers or simplify this to align with the declared narrative design.

Thank you for this important clarification. We agree that our original wording could suggest an unnecessarily large independent data-extraction team for a narrative review. To avoid misunderstanding and to align the Methods with a standard narrative-synthesis workflow, we have revised the text to clarify that two reviewers (KERJ, YMMT) performed all study screening and data extraction using a standardized form. Discrepancies were resolved by discussion and, when needed, with a senior reviewer (DSLD). The additional team members (GGC, COA, EC, GATH, ORL, JJB) contributed to the design of the search strategy, verification of selected data, and methodological oversight, but did not conduct independent data extraction. The updated Methods section (Study selection and Data extraction) now reflects this structure and clearly describes the roles of all contributors.

Conclusion – The final paragraph should be condensed to two or three sentences summarizing the main clinical message and future directions.

We agree and have rewritten the Conclusion as a brief, three-sentence paragraph that summarizes the clinical take-home message (protocolized, multidisciplinary management of HR during CPB) and the main future research priorities (consensus definition, core outcome set, multicenter registries and pragmatic comparative/cost-effectiveness trials).

 Several references appear to derive from non–peer-reviewed or “grey” sources. These should be replaced, where possible, with original peer-reviewed studies to strengthen the scientific rigor of the manuscript. In addition, the reference list contains residual formatting inconsistencies (e.g., spacing, punctuation, capitalization, DOI placement) that need to be standardized.

We have carefully re-examined the reference list and, where possible, replaced grey or non–peer-reviewed sources with original peer-reviewed articles addressing the same topics. References that did not add essential information and could not be adequately substituted have been removed.
In addition, we have systematically revised the reference list to standardize formatting according to the journal’s style (Vancouver), including spacing, punctuation, capitalization, journal abbreviations, and DOI placement.

Once again, we appreciate your thoughtful feedback and believe that these additional changes have further improved the clarity, coherence, and scientific robustness of the manuscript. We hope the revised version meets your expectations.

Sincerely,

Joshuan Barbosa
